# Nitric oxide donor sodium nitroprusside serves as a source of iron supporting *Pseudomonas aeruginosa* growth and biofilm formation

Xavier Bertran i Forga,[1,2] Yaoqin Hong,[2,3] Kathryn E. Fairfull-Smith,[4,5] Jilong Qin,[1,2] Makrina Totsika[1,2]

**ABSTRACT** Biofilm dispersal agents, like nitric oxide (NO), restore antimicrobial effectiveness against biofilm infections by inducing bacteria to shift from a biofilm to a planktonic state, thereby overcoming the antimicrobial tolerance typically associated with biofilms. Sodium nitroprusside (SNP) is a widely used NO donor for investigating the molecular mechanisms underlying NO-mediated biofilm dispersal in the nosocomial pathogen *Pseudomonas aeruginosa*. However, the biofilm effects of SNP are variable depending on the *in vitro* experimental conditions, with some studies reporting enhanced growth in both planktonic and biofilm forms instead of dispersal. These discrepancies suggest that SNP affects *P. aeruginosa* biofilm-residing cells beyond the release of NO. In this study, we compared SNP with another NO donor, Spermine NONOate, to systematically contrast their effects on biofilm and planktonic cultures of *P. aeruginosa*. We found that SNP, but not Spermine NONOate, increased the biomass of *P. aeruginosa* biofilms in microplate cultures. This effect was also observed when biofilm cultures were supplemented with iron. Additionally, supplementation with SNP rescued the planktonic growth of *P. aeruginosa* in iron-depleted media similarly to $FeSO_4$, suggesting that SNP may serve as an iron source. Our findings indicate that the use of SNP as an NO donor in biofilm dispersal may be compromised by its role in promoting both biofilm and planktonic growth through its iron center. Our study cautions investigators using SNP for studying NO-mediated biofilm dispersal.

**IMPORTANCE** Research into biofilm dispersal agent nitric oxide (NO) holds promise for treating biofilm-associated infections. Sodium nitroprusside (SNP), an NO donor widely used in antibiofilm research, has been shown in this study to enhance cell growth and biofilm formation in *Pseudomonas aeruginosa* by acting as a source of iron. Our results suggest that SNP functions both as an NO and an iron donor, with its iron-releasing properties playing a more dominant role in promoting biofilm growth in closed culture systems. This study underscores the dual but conflicting roles of SNP in biofilm growth, which caution its future development as an NO-based therapeutic strategy for biofilm-associated infections.

**KEYWORDS** *Pseudomonas aeruginosa*, biofilm dispersal, nitric oxide, sodium nitroprusside, iron

Address correspondence to Jilong Qin, jilong.qin@qut.edu.au, or Makrina Totsika, makrina.totsika@qut.edu.au.

M.T. is an employee of the GSK group of companies. All other authors declare no conflict of interest.

See the funding table on p. 5.

Biofilms are microbial communities encapsulated in an extracellular polymeric matrix that are ubiquitous in both natural and clinical environments. They cause significant damage through biofouling of equipment and serve as a reservoir for recurrent chronic infections, as well as for food and water contamination (1). In addition, bacteria in biofilms exhibit increased tolerance to antimicrobials and disinfectants compared to their planktonic counterparts (2). Nitric oxide (NO) induces the transition of biofilm

cells to the planktonic state, thereby reducing their antimicrobial tolerance, and thus has been exploited as an antibiofilm agent against clinically and industrially relevant biofilm-forming bacteria (3, 4).

The precise delivery of NO has remained a challenge in both clinical and laboratory settings due to its gaseous and highly reactive nature. This has prompted the development of NO-releasing compounds. Capable of controllably delivering NO, NO-donors have become instrumental in investigating the mechanisms involved in biofilm dispersal. The metal-nitrosyl complex sodium nitroprusside (SNP), an FDA-approved vasodilator, is a model NO donor widely used to induce biofilm dispersal (5–9). SNP consists of a ferrous ($Fe^{2+}$) ion coordinating five cyanide groups and a nitrosonium group ($NO^+$) (Fig. 1A), which is released as NO together with cyanide (10). In flow cells and other open culture systems, SNP successfully disperses *Pseudomonas aeruginosa* biofilms, an effect that has been largely attributed to its NO-releasing properties (5, 11). However, cultures grown in microplates occasionally showed growth increases in planktonic and biofilm bacteria following SNP treatment (12–14). Despite SNP being a well-established NO donor, this underlying effect has remained unexplored. Here, we demonstrate that SNP can provide a source of bioavailable iron for *P. aeruginosa* to support planktonic and biofilm growth.

To explore the effects of SNP on microplate biofilms, we first measured biofilm growth kinetics in minimal media by tracking the biomass of *P. aeruginosa* PAO1 (a model biofilm-forming pathogen) biofilms over 24 h in microtiter plates (Fig. 1B; Text S1). In this system, *P. aeruginosa* biofilms reached maximum biomass at 4 h, followed by a gradual decline, with <10% biofilm biomass remaining after 14 h. Enumeration of live bacterial cells from biofilms at different time points also mirrored this trend (Fig. 1D). As the maximum biomass achieved in microplates occurred at 4 h, 4-h-old biofilms were used to evaluate dispersal effects by two NO donors: SNP and Spermine NONOate (SP-NONOate).

SP-NONOate is an *N*-diazeniumdiolate that spontaneously releases NO in aqueous solutions (12, 15, 16). Consistent with previous reports, SP-NONOate reduced *P. aeruginosa* biomass within 15 minutes (Fig. 1C; Fig. S1A and B) (15). In contrast, SNP induced a dose-dependent biofilm biomass increase after 30 minutes (Fig. 1C; Fig. S1C).

NO has been proposed as a treatment strategy to reduce biofilm-associated antimicrobial tolerance by reverting *P. aeruginosa* biofilm cells to their planktonic state. This hypothesis has been frequently tested using SNP in open culture systems to induce NO-mediated dispersal of *P. aeruginosa* biofilms (5, 7, 8, 12). However, our findings using closed systems contradict this expectation. Rather than dispersing biofilms, we observed that SNP promoted dose-dependent increases in biofilm biomass (Fig. S1C). A previous study using microtiter plates reported that brief exposure to low concentrations of NO correlated with increased *P. aeruginosa* biofilm biomass (16), suggesting that SNP may release NO at levels insufficient to trigger dispersal, and that such low NO concentrations may instead promote biofilm growth. To investigate this tenet, we tested whether low NO concentrations directly promote biofilm growth by treating PAO1 biofilms with a range of sub-dispersing concentrations of SP-NONOate (Fig. S2), but observed no comparable biomass increase. These findings suggest that, in microplates, SNP may enhance biofilm growth through a mechanism independent of NO release.

Iron, which sits at the center of energy metabolism, DNA repair, and cell envelope development, is also an essential micronutrient for *P. aeruginosa* biofilm formation (17). Among the two NO donors, a key structural difference is the presence of an iron center in SNP (Fig. 1A). We reasoned that the increase in biomass with SNP (Fig. 1C) could result from the release of bioavailable iron from SNP. This hypothesis is supported by previous reports showing that *P. aeruginosa* biofilms grown in iron-supplemented media reached higher biomass, and that adding iron to established biofilms induced rapid surface attachment of planktonic cells (15, 18). To test this, we supplemented established *P. aeruginosa* biofilms with matching molecular concentrations of $FeSO_4$, which induced a similar increase in biofilm biomass to that of SNP treatment (Fig. 1C; Fig. S1C). Additionally, media supplementation with SNP or $FeSO_4$ prolonged the biofilm formation phase to 8 h. This stimulated larger accumulations of biofilm biomass containing increased

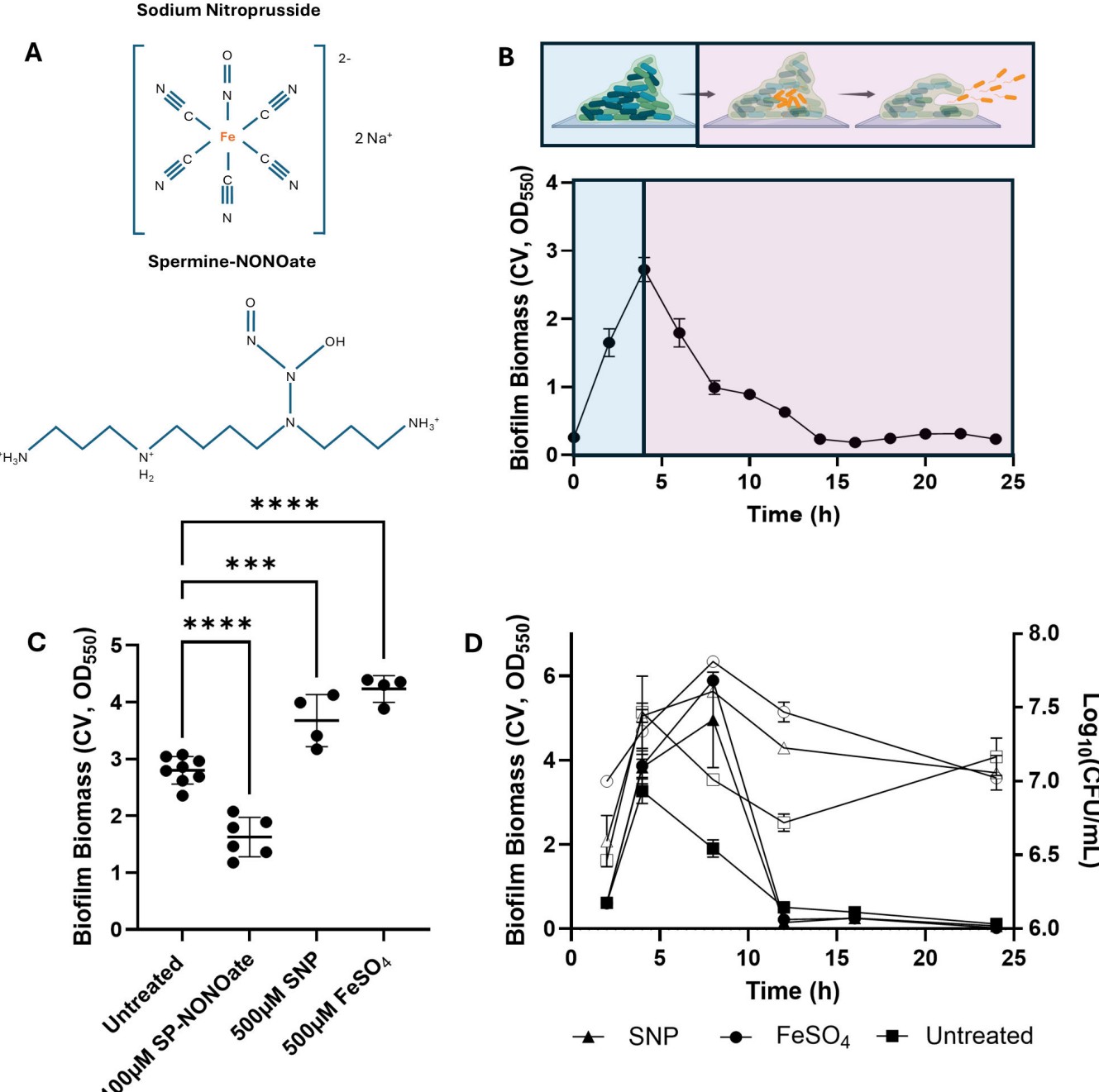

**FIG 1** SNP promotes biomass increase of *P. aeruginosa* PAO1 biofilms. (A) Chemical structure of sodium nitroprusside and Spermine-NONOate. (B) Changes in biofilm biomass over 24 h. $10^7$ CFU/mL bacterial solutions were prepared from overnight cultures using M9 media, transferred to 24-well plates, and incubated at 37°C, shaking. Biofilm biomass was quantified every 2 h by crystal violet staining for up to 24 h. (C) 4 h biofilms were treated with SP-NONOate for 15 min, or FeSO$_4$ or SNP for 30 min, at the indicated concentrations. Data represent at least four independent replicates. (D) 4 h biofilms of either PAO1 WT or *fhp*::Tn were treated with either SP-NONOate or SNP for 15 min or 30 min, respectively, at the indicated concentrations. The data represent two independent replicates. (E) Biofilm cultures were grown in M9 media supplemented with 31.25 µM of SNP or FeSO$_4$. Two independent cultures were included. The means ± SD are represented in the graphs.

biofilm-residing cells (Fig. 1D), whereas SP-NONOate inhibited biofilm biomass accumulation (Fig. S3). These data suggest SNP acts as an iron donor to support biofilm growth.

We next examined whether supplementing iron-chelated M9 medium with SNP could rescue the growth of *P. aeruginosa* planktonic cultures. While standard M9 supported the growth of *P. aeruginosa*, depletion of iron using the iron chelator 2,2′-bipyridyl

significantly inhibited it (Fig. 2; Text S1). However, the addition of SNP reversed this inhibition, affording a comparable restoring effect to that of $FeSO_4$ on the cell growth (Fig. 2). Together, these results suggest that SNP functions as an iron donor, supporting both biofilm and planktonic growth of *P. aeruginosa*.

While the mechanism by which SNP releases iron remains unclear, our data suggest that physiologically relevant iron concentrations become readily available upon SNP addition, overshadowing its NO-related effects in open culture systems. This is evidenced by the rapid increase in biofilm biomass within 30 minutes of SNP exposure, similar to that induced by iron, which was reported to upregulate Psl synthesis and/or enhance attachment of planktonic bacteria (15). Critically, our findings highlight the iron-releasing properties of SNP, a factor that was not fully considered in previous studies investigating its therapeutic potential, likely introducing bias to our understanding of SNP as an NO donor.

The divergent effects of SNP on *P. aeruginosa* biofilms appear to depend on the chosen culture system. Biofilms grown in open flow systems exhibit distinct phenotypes compared to those formed in microplates, where the confined environment allows the accumulation of quorum-sensing signals that alter the metabolism and community-driven behavior of *P. aeruginosa* (19). Our findings, therefore, highlight the utility of microplates as a low-cost, rapid, and high-throughput platform that can reveal treatment effects that may be masked in flow-cell models.

Importantly, NO has been studied in combination with antibiotics to treat biofilm infections (4). SNP has previously been reported to synergize with tobramycin against *P. aeruginosa* biofilms, an antibiotic more effective against metabolically active bacteria (9, 11). Here, we showed that SNP enhances the growth rate of *P. aeruginosa* by donating bioavailable iron, potentially increasing metabolic activity and thereby sensitizing biofilm-embedded bacteria to tobramycin. In addition, iron supplementation has also been shown to enhance the efficacy of antibiotics such as ampicillin, gentamicin, and norfloxacin by promoting the production of reactive oxygen species (20).

Altogether, our results demonstrate that SNP provides a readily available source of iron to growing biofilms of *P. aeruginosa*. Considering most assays studying NO dispersal

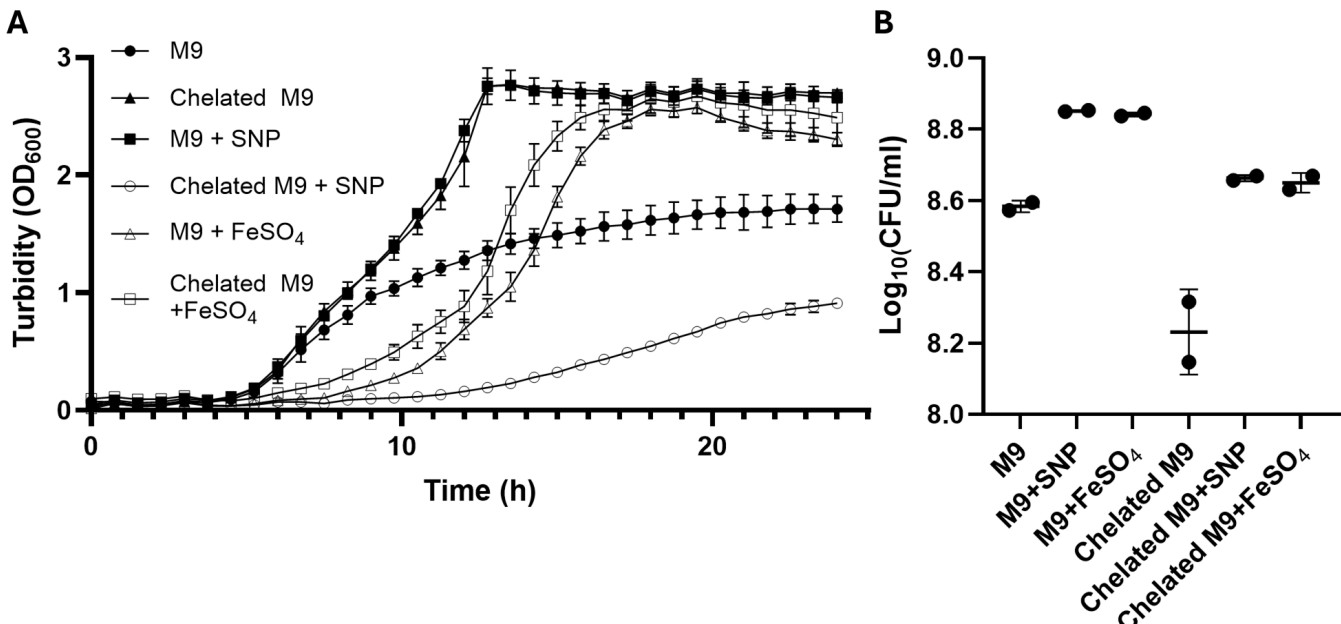

**FIG 2** Growth defects of *P. aeruginosa* PAO1 in iron-depleted media can be rescued by SNP or $FeSO_4$. Initial bacterial cultures were inoculated at an $OD_{600}$ of 0.05 and grown in M9 media containing 500 µM of the iron scavenger 2,2′-bipyridyl. Cultures were supplemented with SNP or $FeSO_4$ under shaking conditions at 37°C in 96-well microtiter plates. (A) Culture optical density was periodically recorded in a plate reader by measuring the absorbance at 600 nm. (B) CFU density of the cultures at 24 h. The data represent two biological replicates. The means ± SD are represented in the graphs.

have been conducted with SNP as the NO donor, our findings should elicit caution when solely attributing reported SNP biofilm responses to NO, as masked secondary effects might have been overlooked.

## ACKNOWLEDGMENTS

This work is funded in part by an Australian Research Council project grant (DP210101317), the Max Planck Queensland Center on the Materials Science of Extracellular Matrices to M.T., and the QUT Amplify Scholarship provided by the Queensland University of Technology (Australia) to X.B.I.F. The Ian Potter Foundation sponsored the CLARIOStar high-performance microplate reader (BMG, Australia). The funders had no role in study design, data collection and analysis, decision to publish, or preparation of the manuscript.

The authors would like to thank Professor Robert EW Hancock (University of Columbia) for providing the *P. aeruginosa* PAO1 strain used in this study.

X.B.I.F. and J.Q. conceptualized the project. X.B.I.F., J.Q., and Y.H. contributed to the experimental design. X.B.I.F. conducted all experiments and contributed to data collection, analysis, and visualization. X.B.I.F., J.Q., Y.H., and M.T. contributed to data interpretation. J.Q. and M.T. supervised the project. K.E.F.-S. and M.T. obtained the funding. X.B.I.F. wrote the original draft, and all authors edited the manuscript.

## AUTHOR AFFILIATIONS

[1]Centre for Immunity and Infection Control, School of Biomedical Sciences, Queensland University of Technology, Brisbane, Queensland, Australia

[2]Max Planck Queensland Centre, Queensland University of Technology, Brisbane, Queensland, Australia

[3]Biomedical Sciences and Molecular Biology, College of Medicine and Dentistry, James Cook University, Douglas, Queensland, Australia

[4]School of Chemistry and Physics, Queensland University of Technology, Brisbane, Queensland, Australia

[5]Centre for Materials Science, Queensland University of Technology, Brisbane, Queensland, Australia

## AUTHOR ORCIDs

Xavier Bertran i Forga http://orcid.org/0000-0002-8374-7470
Yaoqin Hong http://orcid.org/0000-0002-4408-2648
Jilong Qin http://orcid.org/0000-0002-5867-6332
Makrina Totsika http://orcid.org/0000-0003-2468-0293

## FUNDING

| Funder | Grant(s) | Author(s) |
| --- | --- | --- |
| Department of Education and Training | DP210101317 | Makrina Totsika |

## AUTHOR CONTRIBUTIONS

Xavier Bertran i Forga, Conceptualization, Data curation, Formal analysis, Investigation, Methodology, Validation, Visualization, Writing – original draft, Writing – review and editing | Yaoqin Hong, Formal analysis, Methodology, Writing – review and editing | Kathryn E. Fairfull-Smith, Funding acquisition, Writing – review and editing | Jilong Qin, Conceptualization, Formal analysis, Investigation, Methodology, Project administration, Supervision, Validation, Writing – original draft, Writing – review and editing | Makrina Totsika, Formal analysis, Funding acquisition, Project administration, Supervision, Validation, Writing – review and editing

## DATA AVAILABILITY

All data generated or analyzed during this study were included in this article and supplementary files.

## ADDITIONAL FILES

The following material is available online.

### Supplemental Material

**Supplemental material (Spectrum02234-25-s0001.pdf).** Supplemental methods and Fig. S1 to S3.

### Open Peer Review

**PEER REVIEW HISTORY (review-history.pdf).** An accounting of the reviewer comments and feedback.

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
