## [Reviewer comments · Microbiology Spectrum]

Microbiology Spectrum

Nitric oxide donor sodium nitroprusside serves as a source of iron supporting *Pseudomonas aeruginosa* growth and biofilm formation

Xavier Bertran i Forga, Yaoqin Hong, Kathryn E. Fairfull-Smith, Jilong Qin, and Makrina Totsika

Corresponding Author(s): Jilong Qin, Queensland University of Technology

Review Timeline:

Submission Date:	July 23, 2025
Editorial Decision:	August 1, 2025
Revision Received:	August 1, 2025
Accepted:	August 22, 2025

Editor: John Atack

Reviewer(s): The reviewers have opted to remain anonymous.

Transaction Report:

DOI: <https://doi.org/10.1128/spectrum.02234-25>

Re: Spectrum02234-25 (**Nitric oxide donor sodium nitroprusside serves as a source of iron supporting *Pseudomonas aeruginosa* growth and biofilm formation**)

Dear Dr. Jilong Qin:

Thank you for the privilege of reviewing your work. Below you will find my comments.

As per editorial procedure, this manuscript needs to be resubmitted via the Spectrum editorial system so that the submission form can be completed before acceptance. Resubmit the manuscript as you would submit a revision; a new submission is not necessary

Revision Guidelines

Sincerely,
John Attack
Editor
Microbiology Spectrum

Re: Spectrum02234-25R1 (**Nitric oxide donor sodium nitroprusside serves as a source of iron supporting *Pseudomonas aeruginosa* growth and biofilm formation**)

Dear Dr. Jilong Qin:

Your manuscript has been accepted, and I am forwarding it to the ASM production staff for publication. Your paper will first be checked to make sure all elements meet the technical requirements. ASM staff will contact you if anything needs to be revised before copyediting and production can begin. Otherwise, you will be notified when your proofs are ready to be viewed.

Sincerely,
John Attack
Editor
Microbiology Spectrum